# Physiological Effects of Bioactive Compounds Derived from Whole Grains on Cardiovascular and Metabolic Diseases

**Sangwon Chung** [1], **Jin-Taek Hwang** [1,2] **and Soo-Hyun Park** [1,*]

[1] Personalized Diet Research Group, Food Functionality Research Division, Korea Food Research Institute, Wanju-gun 55365, Korea; schung@kfri.re.kr (S.C.); jthwang@kfri.re.kr (J.-T.H.)
[2] Department of Food Biotechnology, University of Science and Technology, Daejeon 34113, Korea
[*] Correspondence: shpark0204@kfri.re.kr

**Abstract:** Cardiovascular diseases are a global health burden with an increasing prevalence. In addition, various metabolic diseases, such as obesity, diabetes, and hypertension are associated with a higher risk of cardiovascular diseases. Dietary strategies based on healthy foods have been suggested for the prevention or improvement of cardiovascular and metabolic diseases. Grains are the most widely consumed food worldwide, and the preventive effects of whole grains (e.g., oats, barley, and buckwheat) on metabolic diseases have been reported. The germ and bran of grains are rich in compounds, including phytochemicals, vitamins, minerals, and dietary fiber, and these compounds are effective in preventing and improving cardiovascular and metabolic diseases. Thus, this review describes the characteristics and functions of bioactive ingredients in whole grains, focusing on mechanisms by which polyphenols, antioxidants, and dietary fiber contribute to cardiovascular and metabolic diseases, based on preclinical and clinical studies. There is clear evidence for the broad preventive and therapeutic effects of whole grains, supporting the value of early dietary intervention.

**Keywords:** whole grain; bioactive compounds; cardiovascular disease; metabolic disease





## 1. Introduction

Human beings are constantly striving for the development of methods for the prevention and treatment of diseases. In the Global Burden of Disease (GBD) 2019 data, the prevalence of cardiovascular disease has approximately doubled over the past 30 years [1]. In addition, cardiovascular diseases are linked to risk factors for metabolic diseases, such as obesity and diabetes [2]. Therefore, lowering the prevalence of metabolic and cardiovascular diseases is considered one of the biggest challenges of today.

It is well known that increases in the prevalence of metabolic and cardiovascular diseases are closely related to lifestyle patterns, such as physical activity and dietary habits [3,4]. Therefore, unlike other diseases, substantial efforts to prevent metabolic and cardiovascular diseases have focused on lifestyle corrections prior to medical treatment. As part of this, food and food ingredients that help prevent metabolic and cardiovascular diseases are attracting attention. In particular, interest in plant-derived phytochemicals is growing [5,6].

The most commonly consumed food in the world is grains (e.g., oats, barley, and buckwheat), which are a source of carbohydrates. In general, people around the world, on average, get almost 50% of their daily energy intake from carbohydrates [7–10]; accordingly, the impact of grain quality on health cannot be overlooked. Previously, many people preferred refined grains with a soft texture. However, interest in whole grains has increased continuously as various nutrients and active ingredients contained in whole grains have been revealed. More than 80% of whole grains consist of endosperm, and the hull contains germ and bran (Figure 1). Most of them consist of carbohydrates, but they also contain proteins, vitamins, and minerals. In particular, germ and bran contain dietary fiber, B vitamins, minerals, and various phytochemicals [11]. However, most of the germ and

bran are removed through the milling process [12]. Many studies have shown that various nutrients and active ingredients in whole grains, especially in the germ or bran, are effective in improving chronic diseases, including metabolic and cardiovascular diseases [13–18]. Representatively, there are phenol- and dietary fiber-derived compounds. In addition, it contains various tocols, which are vitamin E with strong antioxidant activity. They are known to be involved in blood glucose and lipid metabolism, as well as having antioxidant and anti-inflammatory effects [19–22]. Due to the influence of these various bioactive compounds, it has been reported that whole grain consumption is the main factor affecting mortality due to cardiovascular disease [23]. Indeed, three servings of whole grains per day was associated with a reduced risk of mortality from cardiovascular diseases in a meta-analysis [24]. In another meta-analysis study with various cohorts, it was also reported that consuming three servings of whole grains reduced the relative risk of mortality from metabolic diseases and cancers, as well as cardiovascular disease prevalence [25]. However, the associations between various bioactive compounds in whole grains and risk factors for metabolic and cardiovascular diseases have rarely been reviewed.

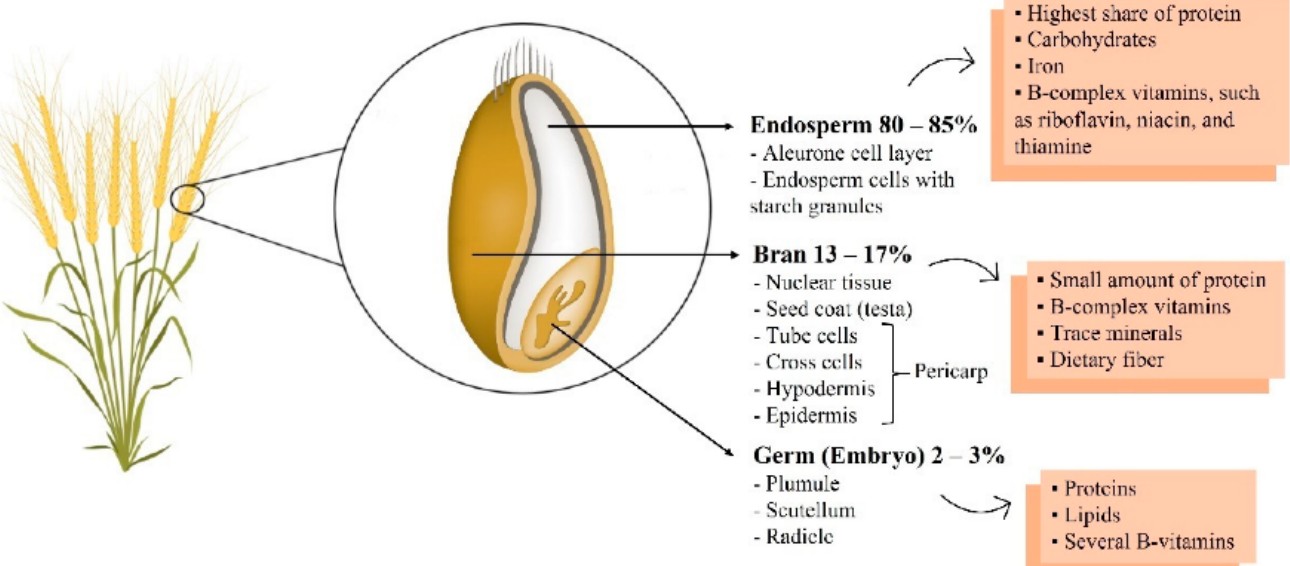

**Figure 1.** Whole grain wheat composition. Reproduced with permission from Sabença et al., Wheat/gluten-related disorders and gluten-free diet misconceptions: A review; published by *Foods*, 2021 [11].

Therefore, a literature review was performed to evaluate the characteristics and functions of active ingredients in whole grains and the effects of whole grain consumption on metabolic and cardiovascular diseases. In particular, we describe recent findings on phenol-, vitamin E-, and dietary fiber-derived bioactive compounds, which are representative bioactive compounds in the following four major groups (including eight compounds): whole grain-specific polyphenols, including alkylresorcinols (ARs), avenanthramides (Avns), ferulic acids (FA), and γ-oryzanol (OZ); flavonoids, including rutin; vitamin E, including tocotrienol and α-tocopherol; dietary fiber, including β-glucan (Figure 2).

**Figure 2.** Main structure of whole grain-derived bioactive compounds (**a**) alkylresorcinol, (**b**) avenanthramide, (**c**) ferulic acid, (**d**) γ-oryzanol, (**e**) rutin, (**f**) tocotrienol, (**g**) α-tocopherol, and (**h**) β-glucan.

## 2. Associations of Whole Grain-Specific Polyphenols with Cardiovascular and Metabolic Diseases

Polyphenols are phenolic compounds with one or more phenol units per molecule and are present in most plant materials, such as vegetables, fruits, and grains [26]. They effectively improve various metabolic and cardiovascular diseases due to their excellent antioxidant and anti-inflammatory effects [27,28]. Representative polyphenol compounds, present in whole grains, are alkylresorcinols (ARs), avenanthramides (Avns), ferulic acids (FA), and γ-oryzanol (OZ) [29–32].

### 2.1. Alkylresorcinols (ARs)

Alkylresorcinols are phenolic lipids, consisting of phenolic rings and long aliphatic hydrocarbon chains [29]. They are mainly present in the bran layer of whole grains and are widely used as biomarkers for the intake of whole grains.

It has been reported in many studies of glucose intolerance, especially in metabolic diseases. According to a prospective cohort study of old adults and pregnant women, blood

AR concentrations and the frequency of whole grain intake were negatively correlated with the incidence of impaired glucose tolerance (IGT) and gestational diabetes mellitus (GDM) [33,34]. In addition, in a case–control study of patients with IGT and type 2 diabetes (T2D), higher levels of DHPPA [3-(3,5-dihydroxyphenyl)-1-propanoic acid], a metabolite of ARs, were correlated with a lower incidence of IGT and T2D [35]. Evidence for the contribution of ARs to glycemic control and the underlying mechanisms has been obtained by studies of mouse models of high-fat, high-sucrose diet (HFHS)-induced obesity and glucose intolerance. In this model, ARs alleviate glucose intolerance and insulin resistance by decreasing insulin and leptin secretion and increasing insulin-stimulated hepatic serine/threonine protein kinase B phosphorylation. In addition, cholesterol levels decreased via the upregulation of hepatic cholesterol synthetic genes, such as sterol regulatory element binding transcription factor 2 (*Srebf2*) and 3-hydroxy-3-methylglutaryl-CoA synthase 1 (*Hmgcs1*), as well as increasing fecal excretion of cholesterol [36]. As such, ARs showed an excellent effect in controlling blood glucose and lipids that directly or indirectly affect cardiovascular diseases. Therefore, these study results provide scientific evidences to support a human study that elevated blood DHPPA levels, induced by whole grain intake, lowered the prevalence of ischemic stroke [16].

## 2.2. Avenanthramides (Avns)

Avenanthramides are *N*-cinnamoyl derivatives of anthranilic acid, a group of phenolic alkaloids [30]. They exist in various plant materials, but they are widely known as phytochemicals in whole grain oats.

It has been reported to prevent vascular diseases in various preclinical studies. In LDLr$^{-/-}$ mice fed a high-fat diet (HFD), oat bran with Avns reduced total cholesterol (TC) and atheroma lesions in the aortic valve. These effects were more pronounced in the high-Avns diet group than in the low-Avns diet group [17]. The mechanism, for the protective effects of Avns against vascular diseases, has been investigated using vascular endothelial and smooth muscle cells (SMCs). Avns inhibit the proliferation of SMCs by increasing the levels of nitric oxide (NO), cyclic guanosine monophosphate (cGMP), upregulating endothelial nitric oxide synthase (eNOS), and protein kinase B (Akt) in human umbilical vein endothelial cells (HUVECs), human aortic endothelial cells (HAECs), and SMCs [37,38]. Nie et al. [39] reported that Avns induce G1 phase arrest of SMC proliferation in rat embryonic aortic SMCs by suppressing the phosphorylation of retinoblastoma protein (pRb), involved in the cell cycle. The phosphorylation of pRb and cell cycle arrest are induced by the inhibition of cyclin D1 expression and upregulation of the p53-p21cip pathway. Since vascular inflammation is one of the factors that induces cardiovascular diseases, including atherosclerosis, its effects have also been explored in various in vitro models. In IL-1β- or TNF-α-stimulated endothelial cells or SMCs, the secretion of inflammatory mediators is increased by mitogen-activated protein kinase (MAPK) signaling, and cell migration and proliferation are increased by matrix metalloproteinase (MMP) activity. Avns decrease the secretion of pro-inflammatory cytokines and inhibit the migration and proliferation of SMCs by downregulating MAPK signaling and suppressing MMP protein expression and NF-κB activation [40–42].

## 2.3. Ferulic Acid (FA) and γ-Oryzanol (OZ)

Ferulic acid (4-hydroxy-3-methoxycinnamic acid) is a derivative of hydroxycinnamic acid, a type of phenolic acid [31]. γ-oryzanol is a ferulic acid mixture in which triterpenoids are bound to esterified ferulic acid [43]. Both compounds are representative bioactive substances of whole-grain rice bran [31,32]. FA and OZ are effective in improving cardiovascular and metabolic diseases.

The beneficial effects of FA and OZ on cardiovascular disease have been confirmed in animal models fed a HFD or high-fat, high-fructose diet (HFHF). In this model, FA and OZ decreased body weight, body fat, and blood lipid (TC, triglyceride [TG], low-density lipoprotein cholesterol [LDL-C], free fatty acid [FFA]) levels and increased high-density

lipoprotein cholesterol (HDL-C). It also inhibited the accumulation of TC and TG in the liver and aorta and improved markers of oxidative stress [glutathione peroxidase (GSH-Px), catalase (CAT), glutathione reductase (GR), paraoxonase (PON1), total antioxidant capacity (TAC), malondialdehyde (MDA), lipid peroxide, vitamin E] and inflammation [tumor necrosis factor-$\alpha$ (TNF-$\alpha$), interleukin -6 (IL-6), C-reactive protein (CRP)] [15,18,44–48]. In particular, OZ increased the fecal excretion of TC and inhibited lipid accumulation [46,47]. In HepG2 cells, FA and OZ suppressed lipid accumulation by inhibiting the expression of fatty acid synthase (FAS), acetyl-CoA carboxylase (ACC), sterol regulatory element binding protein 2 (SREBP-2), 3-hydroxy-3-methylglutaryl coenzyme A reductase (HMGCR), and diacylglycerol *O*-acyltransferase 1 (DGAT1) [18,48].

Furthermore, the effects of FA and OZ on diabetes have been examined in various animal models. FA and OZ alleviate glucose intolerance and insulin resistance by decreasing glucose, glucose area under the curve (AUC), insulin, and homeostatic model assessment for insulin resistance (HOMA-IR) in animal models fed a HFHF diet or a high-carbohydrate, high-fat diet (HCHF) [15,18,48]. Blood glucose level and glucose intolerance were improved in *db/db* mice and in a streptozotocin-induced diabetes rat model [45,49]. These effects are presumed to be induced by FA and OZ, which activate glucokinase activity and increase hepatic glycogen synthesis by elevating AMP-activated protein kinase (AMPK) and Akt phosphorylation in the liver [45,48].

FA has also been reported to regulate blood pressure. In HUVECs, FA treatment increased NO and cGMP levels by upregulating Akt1 and eNOS expression and decreasing superoxide levels [38]. In spontaneously hypertensive rats (SHRs), FA inhibits the activity of an angiotensin-converting enzyme (ACE), which is responsible for the conversion of the inactive form of angiotensin I to the active vasoconstrictor angiotensin II [50]. In addition, FA improved NO bioavailability by reducing oxidative stress and the production of thromboxane $B_2$ (TXB$_2$) [15,51]. As a result, it was confirmed that FA has an anti-hypertensive effect by suppressing vasoconstriction via regulation of renin-angiotensin system (RAS) and endothelial function.

In addition, FA and OZ reduce the thickness of arterial blood vessel walls in animal models; therefore, they are expected to prevent cardiovascular diseases and to have hypolipidemic, hypoglycemic, and blood pressure-lowering effects [15] (Table 1).

**Table 1.** Effects of phenolic compounds on cardiovascular and metabolic diseases.

| Bioactive Compounds | Food Source | Study Subject or Model | Treatment or Intervention | Results | References |
|---|---|---|---|---|---|
| | | | Alkylresorcinols (ARs) | | |
| ARs | Whole grain | Human study<br>- Nested case control study<br>- 64 years, women | - 5 years follow-up<br>- concentration of ARs | - Concentration of AR homologs C17 and C19 lower in subjects with IGT than that in normal subjects | [33] |
| ARs | Whole grain | Human study<br>- Prospective cohort study<br>- Pregnant women (gestational weeks 11–14) | - follow-up during pregnancy<br>- total plasma AR Q1 (median 66 nmol/L) vs. Q4 (median 706 nmol/L)<br>- weekly whole grain consumption Q1 (median 1.2 times/week) vs. Q4 (median 14.5 times/week) | - Frequency of whole-grain consumption lower in subjects with GDM than in subjects without GDM<br>- Median concentration of ARs lower in subjects with GDM than in subjects without GDM<br>- Highest ARs concentration quartile with an RR of 0.50 compared to lowest quartile | [34] |
| DHPPA [3-(3,5-dihydroxyphenyl)-1-propanoic acid] | Whole grain | Human study<br>- Case-control study<br>- Subjects with T2D and IGT<br>- ≥30 years | - plasma DHPPA concentrations Q1 (<6.56 nmol/L) vs. Q4 (≥17.98 nmol/L) | - Negatively correlated with DHPPA concentrations and odds of T2D and IGT | [35] |
| ARs | Wheat bran | In vivo<br>- 3-week-old, C57BL/6J mice fed with HFHS | - 10 weeks<br>- 0.4% ARs containing diet | - Suppressed increases in bodyweight and hepatic TG accumulation<br>- Suppressed increases in blood insulin and leptin concentration<br>- Reduced levels of fasting and postprandial glucose and improved glucose intolerance and insulin resistance<br>- Increased phosphorylation of insulin-stimulated hepatic serine/threonine protein kinase B<br>- Increased fecal excretion of cholesterol and reduced blood cholesterol concentration<br>- upregulated expression of hepatic cholesterol synthetic genes (Srebf2, Hmgcs1) | [36] |
| DHPPA (3-(3,5-dihydroxyphenyl)-1-propanoic acid) | Whole grain | Human study<br>- Case-control study<br>- Subjects with ischemic stroke<br>- ≥35 years | - Concentration of DHPPA | - Negatively correlated with DHPPA concentrations and odds of ischemic stroke | [16] |

Table 1. *Cont.*

| Bioactive Compounds | Food Source | Study Subject or Model | Treatment or Intervention | Results | References |
|---|---|---|---|---|---|
| | | | Avenanthramides (Avns) | | |
| Avns | Oats | In vivo - 5-week-old, Ldlr$^{-/-}$ mice fed with HFD | - 16 weeks - oat bran with low concentrations of Avns (8.8 g/kg, HFLA) or oat bran with high concentrations of Avns (480 g/kg, HFHA) diet | - Both oat-based diets decreased HFD-induced atheroma lesions in the aortic valve - HFHA administration for atheroma lesions is more effective than HFLA - TC level similarly reduced in both oat-based diets | [17] |
| Avn-A, B, C | Oats | In vitro - HUVECs | - Avn-A, B, C 1 μM | - Increased NO and cGMP levels - Increased Akt1 and eNOS expression | [38] |
| Avn-c | Oats | In vitro - Human aortic SMCs and HAECs | - Avn-c 40, 80, 120 μM | - Inhibited SMC proliferation - Increased NO secretion in both SMC and HAEC - Upregulated eNOS mRNA expression in both SMC and HAEC | [37] |
| Avn-c | Oats | In vitro - A10 rat embryonic aortic SMCs | - Avn-c 40, 80, 120 μM | - Arrested SMC proliferation at G1 phase - Decreased phosphorylation of pRb - Decreased cyclin D1 expression - Increased cyclin-dependent kinase inhibitor p21cip1 expression - Increased expression and stability of p53 protein | [39] |
| Avns | Oats | In vitro - IL-1β-stimulated HAECs | - Avns 4, 20, 40 ng/mL | - Decreased U937 monocytic cells adhesion in IL-1β-stimulated HAEC - Suppressed expressions of ICAM-1, VCAM-1, and E-selectin - Suppressed secretion of proinflammatory cytokines IL-6, IL-8, and MCP-1 | [40] |
| Avn-c | Oats | In vitro - TNF-α-stimulated HASMCs | - Avn-c 50, 100 ng/mL | - Inhibited cell migration - Suppressed increasing MMP-9 protein and mRNA levels - Inhibited MMP-9 enzyme activity - Reduced IL-6 level - Suppressed nuclear protein translocation of nuclear factor kappa B (NF-κB) - Reduced expression of ERK, JNK, and p38 phosphorylation | [41] |

| Bioactive Compounds | Food Source | Study Subject or Model | Treatment or Intervention | Results | References |
|---|---|---|---|---|---|
| Avns | Oats | In vitro<br>- IL-1β-stimulated HAECs<br>- IL-1β-stimulated HUVECs | - AvnsO (oat extract) 4, 20, 40 mg/mL<br>- Avn-c (Avn-c) 1, 10, 20, 40, 100 μM<br>- CH3-Avn-c (methyl ester of Avn-c) 1, 10, 40, 100 μM | - AvnsO, Avn-c, and CH3-Avn-c suppressed NF-κB p50 DNA binding activity<br>- CH3-Avn-c decreased expression of IL-6, IL-8, and MCP-1<br>- AvnsO, Avn-c, and CH3-Avn-c inhibited NF-κB-dependent reporter gene expression activated by TNFR-associated factor 2 and 6 (TRAF2, TRAF6) and NF-κB-inducing kinase<br>- Decreased phosphorylation of IKKβ and IκB and stabilized IκB protein | [42] |
| Ferulic acid (FA) and γ-oryzanol (OZ) | | | | | |
| FA and OZ | Rice bran | In vivo<br>- 4-week-old, C57BL/6 mice fed with HFD | - 7 weeks<br>- 0.5% FA or OZ containing diet | - Decreased bodyweight gain in both groups<br>- Decreased blood TG, TC, and increased plasma HDL-C in both groups<br>- Decreased hepatic TG and TC in both groups<br>- Decreased ME and FAS in both groups—increased GSH-Px, CAT, GR, and PON1 in both groups<br>- Increased fecal excretion of TC in OZ treated group | [46] |
| FA and OZ | Rice bran | In vivo<br>- $F_1B$ Golden Syrian hamsters fed with HCD | - 10 weeks<br>- 0.5% FA or OZ containing diet | - Decreased blood TC, and non-HDL-C in both groups<br>- Decreased plasma lipid hydroperoxides in both groups<br>- Decreased aortic TC, and TG accumulation in both groups<br>- Decreased blood TG and increased HDL-C in OZ treated group<br>- Increased fecal excretion of TC and coprostanol in OZ treated group<br>- Increased plasma vitamin E in FA treated group | [47] |

**Table 1.** *Cont.*

| Bioactive Compounds | Food Source | Study Subject or Model | Treatment or Intervention | Results | References |
|---|---|---|---|---|---|
| FA and OZ | Whole grain wheat | In vivo<br>- SD rats (weighing 180–200 g) fed with HFFD | In vivo<br>- 16 weeks<br>- 0.24% FA or 0.71% OZ containing diet | In vivo<br>- Decreased body weight in both groups<br>- Decreased blood TG, TC, LDL-C, and increased HDL-C in both groups<br>- Decreased fasting blood glucose and HOMA-IR in both groups<br>- Decreased glucose AUC in FA treated group<br>- Increased AMPK and Akt phosphorylation in liver tissues in both groups | [48] |
| | | In vitro<br>- HepG2 cell line<br>- FFA induced lipid/glucose metabolic dysfunction cell model | In vitro<br>- FA or OZ 100 µM | In vitro<br>- Decreased TG accumulation in both groups<br>- Decreased MDA level in FA treated group<br>- Decreased AMPK phosphorylation and increased Akt phosphorylation in both groups<br>- Decreased DGAT1 expression in both groups | |
| FA and OZ | Rice bran | In vivo<br>- SD rats (weighing 350–360 g) with HFFD | In vivo<br>- 13 weeks<br>- 0.05% FA or 0.16% OZ containing diet | In vivo<br>- Decreased body weight and fat index in both groups<br>- Decreased blood TC, TG, LDL-C, and FFA in both groups<br>- Decreased hepatic TC in both groups<br>- Decreased glucose, insulin, glucose AUC, and HOMA-IR in both groups<br>- Increased TAC and decreased MDA in both groups<br>- Decreased CRP and IL-6 and increased adiponectin in OZ treated group<br>- Decreased TNF-α in FA treated group | [18] |
| | | In vitro<br>- HepG2 cell line | In vitro<br>- FA and OZ 50 mM | In vitro<br>- Decreased TG accumulation in OZ treated group<br>- Decreased FAS, ACC, SREBP-2, HMGCR expression in both groups<br>- Decreased stearoyl coA desaturase-1 expression in OZ treated group | |

**Table 1.** *Cont.*

| Bioactive Compounds | Food Source | Study Subject or Model | Treatment or Intervention | Results | References |
|---|---|---|---|---|---|
| Phenolic acid fraction (ethyl acetate fraction, EAE) and FA | Rice bran | In vivo<br>- 8-week-old, C57BL/KsJ-db/db mice | - 17 days<br>- EAE 0.2 g/kg or FA 0.05 g/kg<br>- oral administration | - Decreased glucose and increased insulin<br>- Increased hepatic glycogen synthesis and glucokinase activity<br>- Decreased TC and LDL-C | [45] |
| FA | Whole grain and plant materials | In vivo<br>- STZ-induced diabetic rats (weighing 160–170 g) | - 45 days<br>- FA 10, 40 mg/kg<br>- intragastric intubation | - Decreased blood glucose<br>- Decreased blood FFA, TG, TC, and phospholipids | [49] |
| FA | Rice bran and plant materials | In vivo<br>- SD rats (weighing 220–250 g) fed with HCHF diet and 15% fructose in drinking water | - 6 weeks<br>- FA 30, 60 mg/kg<br>- oral administration | - Decreased fasting blood glucose, glucose AUC, HOMA-IR, TC, TG and increased HDL-C<br>- Decreased SBP, DBP, MAP, HR, HVR and increased HBF<br>- Decreased plasma MDA and $p47^{phox}$ protein expression<br>- Increased nitrate/nitrite levels and eNOS protein expression<br>- Decreased mesenteric arterial wall thickness, media to lumen ratio, cross-sectional area of the media layer, and lumen area | [15] |
| FA | Whole grain oats | In vitro<br>- HUVECs | - FA, FA derivatives (isoferulic acid, hydroferulic acid, ferulic acid 4-O-glucuronide, isoferulic acid 3-O-sulfate, dihydroferulic acid 4-O-glucuronide) 1 μM | - Increased NO and cGMP levels<br>- Decreased superoxide levels<br>- Increased Akt1 and eNOS expression | [38] |
| FA | Plant materials | Ex vivo<br>- 20 to 24-week-old, SHRs | - FA 0.01, 0.1, 1 mmol/ml | - Relaxed phenylephrine-induced contraction<br>- Relaxation partially supressed by removing endothelium or by pretreatment with L-NAME<br>- Decreased $TXB_2$ production<br>- Decreased NADPH-dependent superoxide anion levels<br>- Increased ACh-induced vascular relaxation | [51] |

**Table 1.** *Cont.*

| Bioactive Compounds | Food Source | Study Subject or Model | Treatment or Intervention | Results | References |
|---|---|---|---|---|---|
| FA | Wheat bran and plant materials | In vivo - 6-week-old, SHRs | - single dose - FA 9.5 mg/kg - oral administration | - Decreased SBP - Decreased blood TC, TG, ACE activity | [50] |

ACC, acetyl-CoA carboxylase; ACE, angiotensin converting enzyme; ACh, acetylcholine; Akt, protein kinase B; AMPK, AMP-activated protein kinase; AUC, area under the curve; CAT, catalase; cGMP, cyclic guanosine monophosphate; CRP, C-reactive protein; DBP, diastolic blood pressure; DGAT1, diacylglycerol O-acyltransferase 1; eNOS, endothelial nitric oxide synthase; ERK, extracellular signal regulated kinase; FAS, fatty acid synthase; FFA, free fatty acid; GDM, gestational diabetes mellitus; GR, glutathione reductase;GSH-Px, glutathione peroxidase; HAECs, human aortic endothelial cells; HASMCs, human arterial smooth-muscle cells; HBF, hindlimb blood flow; HCD, hypercholesterolemic diet; HCHF, high-carbohydrate and high-fat; HDL-C, high-density lipoprotein cholesterol; HepG2, human hepatocellular carcinoma; HFD, high-fat diet; HFFD, high-fat and high-fructose diet; HFHA, high fat containing regular oat brans with high levels of Avns; HFHS, high-fat and high-sucrose diet; HFLA, high fat containing regular oat brans with low levels of Avns; HMGCR, 3-hydroxy-3-methylglutaryl coenzyme A reductase; Hmgcs1, 3-hydroxy-3-methylglutaryl-CoA synthase 1; HOMA-IR, homeostatic model assessment for insulin resistance; HR, heart rate; HUVECs, primary human umbilical vein endothelial cells; HVR, hindlimb vascular resistance; ICAM-1, intercellular adhesion molecule-1; IGT, impaired glucose tolerance; Q, Quartile; IKKβ, IκB kinase; IL, interleukin; IκB, inhibitor of κB; JNK, c-Jun N-terminal kinase; LDL-C, low density lipoprotein cholesterol; L-NAME, L-NG-Nitro arginine methyl ester; MAP, mean arterial pressure; MCP-1, monocyte chemoattractant protein-1; MDA, malondialdehyde; ME, malic enzyme; MMP-9, matrix metallopeptidase-9; NADPH, Nicotinamide adenine dinucleotide phosphate; NF-κB, nuclear protein translocation of nuclear factor kappa B; NO, nitric oxide; PON1, paraoxonase 1; pRb, retinoblastoma protein; RR, risk ratio; SBP, systolic blood pressure; SD, Sprague-Dawley; SHRs, spontaneously hypertensive rats; SMCs, smooth muscle cells; Srebf2, sterol regulatory element binding transcription factor 2; SREBP, sterol regulatory element binding protein 2; STZ, streptozotocin; T2D, type 2 diabetes; TAC, total antioxidant capacity; TC, total cholesterol; TG, triglyceride; TNFR, tumor necrosis factor receptor; TNF-α, tumor necrosis factor alpha; TRAF, tumor necrosis factor receptor-associated factor; TXB2, thromboxane B2; VCAM-1, vascular cell adhesion molecule-1.

### 3. Associations of Flavonoids with Cardiovascular and Metabolic Diseases

Flavonoids, a subclass of phytochemicals generally originating from plant foods, can be classified into six major groups (i.e., flavanones, flavones, flavonols, flavan-3-ols, anthocyanins, and isoflavones), each of which includes various bioactive compounds [52].

The associations between flavonoids from whole grains and cardiovascular and metabolic diseases have also been demonstrated. According to Hu et al., the flavonoid fraction from tartary buckwheat has favorable effects on insulin and glucose metabolism in mice fed 20% high fructose water, in a dose-dependent manner, at flavonoid fraction concentrations of 200, 400, and 800 mg/kg bw. The mice fed the flavonoid fraction from tartary buckwheat showed decreases in serum glucose and HOMA-IR, along with TC, LDL-C, and TG, which were induced by high fructose. In particular, the expression of insulin signaling pathway-related proteins in the liver, including insulin receptor substrate-1 (IRS1) phosphorylation, phosphoinositide 3-kinases (PI3K) (p85), protein kinase B (Akt) phosphorylation, and translocation of glucose transporter type 4 (GLUT4), increased [53]. In another study, the flavonoid fraction from tartary buckwheat improved hypertension parameters by improving insulin sensitivity in SHRs, compared to that in normal rats. Treatment with the tartary buckwheat flavonoid fraction, at a dose of 100 mg/kg/day, reduced the level of systolic blood pressure (SBP) in SHRs. In addition, treatment with tartary buckwheat flavonoid fractions increased the vasodilator response to insulin and reduced the phosphorylation of IRS1 with an improvement in antioxidative stress [54].

*Rutin*

Rutin, which is also called quercetin-3-*O*-rutinoside, is a flavonoid glycoside found in buckwheat, citrus fruits, onions, wine, and grapes. Quercetin is the well-known aglycone of rutin. The biological activities of rutin, such as its hypoglycemic, hypolipidemic, hepatoprotective, and anti-inflammatory effects, have been demonstrated in several previous studies [55,56].

It has been reported that rutin and epigallocatechin gallate (EGCG) from buckwheat improved insulin metabolism in an in vitro model. According to Cai and Lin [57], rutin and EGCG preserve insulin signaling and regulate lipogenesis. Rutin stimulated IRS2 signaling in pancreatic β-cells and suppressed glucolipotoxic effects, via the AMPK signaling activation, to inhibit lipogenesis-related enzyme activity, and EGCG had similar effects. In addition, female adults who ate buckwheat cookies rich in rutin showed lower levels of HDL-C and TC than those in individuals who ate common buckwheat cookies, which are low in rutin, in a clinical trial [58] (Table 2).

**Table 2.** Effects of flavonoids on cardiovascular and metabolic diseases.

| Bioactive Compounds | Food Source | Study Subject or Model | Treatment or Intervention | Results | References |
|---|---|---|---|---|---|
| Flavonoid fraction |||||| 
| Flavonoid fraction | Tartary buckwheat | In vivo - Mice fed high fructose in drinking water (weighing 18–22 g) | - 8 weeks - Tartary buckwheat flavonoids 200, 400, and 800 mg/kg/day in drinking water | - Improved insulin sensitivity and glucose tolerance - Reversed and attenuated of insulin action on IRS1 phosphorylation, Akt and PI3K, and GLUT4 translocation in insulin-resistant liver | [53] |
| Flavonoid fraction | Tartary buckwheat | In vivo - 6-week-old, SHRs | - 8 weeks - Tartary buckwheat flavonoids 100 mg/kg/day - oral administration | - Reduced SBP - Increased vasodilator response to insulin and reduced IRS-1 phosphorylation at serine 307 - Attenuated hypertension development by reducing vascular oxidative stress | [54] |
| Rutin ||||||
| Rutin | N/A | In vitro - Insulinoma pancreatic β cells of RIN-m5F rat | - EGCG or buckwheat flavonoid Rutin 0.1, 10 μM | - Stimulated IRS2 metabolic pathway in pancreatic β cells of rat - Activated AMPK pathway to suppress activation of lipogenic enzyme | [55] |
| Rutin | Buckwheat cookies | Human study - RCT, crossover - Female adults - mean age 46 years | - 2 weeks - Common buckwheat (low rutin content, 16.5 mg rutin equivalents/day) vs. Tartary buckwheat (high rutin content, 359.7 mg rutin equivalents/day) cookies | - Reduced total serum cholesterol and HDL-C and improved lung vital capacity - Improved MPO, an indicator of inflammation | [58] |

AMPK, AMP-activated protein kinase; EGCG, epigallocatechin gallate; GLUT4, glucose transporter type 4; HDL-C, high-density lipoprotein cholesterol; IRS, insulin receptor substrate; MPO, myeloperoxidase; PI3K, phosphoinositide 3-kinases; RCT, randomized, controlled clinical trial; SHRs, spontaneously hypertensive rats.

## 4. Associations of Vitamin E with Cardiovascular and Metabolic Diseases

According to Halliwell and Gutteridge [59], an antioxidant is defined as any substance that significantly suppresses or delays the oxidation of the substrate when existing at lower level than the oxidizable substrate. In other words, antioxidants are able to defend against oxidative damage caused by free radicals [60]. Thus, antioxidants, such as vitamin E, contribute to cardiovascular and metabolic disease prevention by the management of oxidation-related metabolism, such as oxidative stress. Oxidative stress is linked to diabetes risk factors as well as cardiovascular risk factors, such as hyperlipidemia, hypertension, endothelial dysfunction, and vascular inflammation [61–63]. Indeed, observational studies have found a negative relationship between stroke, coronary heart disease, and mortality and dietary intake or blood levels of antioxidants [64,65].

### 4.1. Tocotrienol

Tocotrienol belongs to the vitamin E group and consists of a hydrophobic carbon side chain and a 6-chromanol ring with three unsaturated bonds. Tocotrienol is more incorporated in the lipid membrane than tocopherols [66].

It has been observed that the tocotrienol-rich fraction of rice bran oil (RBO) has protective effects on glucose and lipid metabolism. In an experimental study, the administration of tocotrienol-rich fractions (TRF), for 6 weeks, decreased the level of TC, LDL-C, TG, apolipoprotein B, and glucose in hypercholesterolemic swine. In addition, tocotrienol in the blood maintained low serum lipid levels, indicating that the alteration of tocotrienols to tocopherols is not fast [67].

In hyperlipidemic albino male rats fed an atherogenic diet, TRF, derived from RBO at concentrations of 0, 4, 8, 12, 25, or 50 mg TRF/kg bw/day, reduced plasma TG, TC, and LDL-C. During TRF treatment, the activity of HMG-CoA reductase also decreased [68].

In diabetic rats, the TRF of RBO and palm oil (PO), administered at a dose of 200 mg/kg bw/day for 8 weeks, showed hypoglycemic effects by lowering fasting blood glucose and HbA1c levels. However, blood glucose was lowered more substantially by treatment with PO-TRF than with RBO-TRF, presumably because RBO has a lower bioavailability due to the higher concentration of tocopherol, which might reduce the bioavailability of tocotrienol. In addition, TRF supplementation attenuated renal dysfunction and restored the antioxidant status in diabetic rats [69]. In another in vivo study of a diabetic model, supplement with TRF of RBO and PO improved the glycemic status and lipid parameters. Supplementation with TRF of RBO, at a dose of 400 mg/kg bw/day, and palm oil, at a dose of 200 mg/kg bw/day, reduced HbA1c, blood glucose, VLDL-C, LDL-C, TC, and TG levels compared to the corresponding levels in diabetic control mice. The levels of antioxidant enzymes decreased in the kidneys of diabetic mice groups, but these decreases were attenuated by treatment with RBO and PO-TRF [70]. In addition, TRF, isolated from RBO at a dose of 10 mg/kg bw/day, decreased LDL-C and TC by 30% and 67%, respectively, decreased alkaline phosphatase (ALP) activity, and maintained low activity of glutathione-S-transferase (GST) in the mammary glands and liver of carcinogenic and hypercholesterolemic rats induced by 7,12-dimethylbenz [α]anthracene (DMBA) [71]. Tocotrienols and bran suppressed body weight gain induced by a HFD, regardless of brain oxidation or serum lipid levels [72].

Tocotrienols, from heated and stabilized rice bran with a healthy diet, also improved serum lipid profiles in those who have hypercholesterolemia. The subjects were divided into three phases and consumed their typical diet, the American Heart Association Step-1 diet, which was limited to fat intake less than 30% energy and 300 mg/day cholesterol intake, as well as the same diet with capsules of 25 to 200 mg/day of TRF in sequence for 35 days each. As a result, a dose of 100 mg/day TRF decreased serum levels of TG, TC, LDL-C, and apolipoprotein B [14]. In another clinical trial, treatment with tocotrienol for 60 days decreased serum LDL-C and TC levels in subjects with T2D and hyperlipidemia [73]. These findings suggest that tocotrienols have preventive effects on coronary heart disease, hyperlipidemia, and atherogenesis [14,73].

### 4.2. α-Tocopherol

α-Tocopherol is a liposoluble stereoisomer of vitamin E compounds. It has a chromanol ring and a 16-carbon phytyl side chain that is saturated [74]. These structures confer lipophilicity, which means the properties of chemical components dissolve in lipids, to lipid bilayers of lipoproteins or membranes [74]. Tocopherols are rich in various types of foods, while tocotrienols can be obtained from rice bran and PO [75,76].

Previous studies have shown that they have preventive effects against cardiovascular disease [77]. However, α-tocopherols inhibit the beneficial effects of tocotrienols on metabolic disorders [69]. For instance, it was reported that treatment with α-tocopherol attenuated the TG- and TC-lowering effects of tocotrienol from rice bran in F344 rats, via the suppression of the lipid metabolism-related genes CPT-1a andCyp7a1, whereas treatment with α-tocopherol alone did not show lipid parameter-lowering effects [78]. A comparative analysis of the effects of tocotrienol and α-tocopherol in an in vivo rice bran model has also shown that anti-hypertensive effects were greater in the tocotrienol treatment group than in the α-tocopherol treatment group [79]. Hence, the administration of rice bran extract containing phytosterol, tocotrienol, and α-tocopherol did not alter body weight, lipid levels, or glucose metabolism, but it attenuated the increase in MDA in obese diabetic mice. In addition, the level of α-tocopherol in obese diabetic mice fed rice bran extract was higher than that in obese diabetic mice fed a normal diet [80] (Table 3).

Table 3. Effects of vitamin E on cardiovascular and metabolic diseases.

| Bioactive Compounds | Food Source | Study Subject or Model | Treatment or Intervention | Results | References |
|---|---|---|---|---|---|
| | | | Tocotrienol | | |
| TRF | N/A | In vivo<br>- 4-month-old, Hypercholesterolemic swine | - 6 weeks<br>- Corn-soybean control diet vs. control diet added with 50 µg of d-P21-tocotrienol, d-P25- tocotrienol, γ-tocotrienol, or TRF | - Reduced serum TG, LDL-C, TC, and apolipoprotein B<br>- Reduced glucose level<br>- Level of insulin greater in treatment group | [67] |
| TRF | RBO | In vivo<br>- Hyperlipidemic albino male rats fed with atherogenic diet (weighing 175–200 g) | - 1 week<br>- TRF derived from rice bran oil with concentrations of 0, 4, 8, 12, 25 or 50 mg TRF/kg bw/day<br>- mouth intubation | - Decreased lipid parameters including TC, LDL-C, and TG in a dose-dependent manner<br>- Attenuated HMG-CoA reductase activity | [68] |
| TRF | RBO, PO | In vivo<br>- 9 to 10-week-old, STZ-induced diabetic rats (weighing 250 g) | - 8 weeks<br>- PO and RBO derived TRF at dose of 200 mg/kg bw/day by gavage | - Lowered blood glucose and HbA1c levels<br>- Decreased protein in urine, serum NO, TBARS, and MDA level and increased SOD and catalase level | [69] |
| TRF | RBO, PO | In vivo<br>- T2D rats (weighing 175–200g) | - 16 weeks<br>- Supplementation with TRF of RBO at a dose of 400 mg/kg bw/day and PO at a dose of 200 mg/kg bw/day by gavage | - Improved glycemic status and lipid parameters by reducing blood glucose, HbA1c, VLDL-C, LDL-C, TC, and TG levels<br>- Ameliorated lipid induced nephropathy by its anti-hyperglycemic, anti-hyperlipidemic, and activities of antioxidant as well as by modulation of TGF-β | [70] |
| TRF | RBO | In vivo<br>- Rats treated with the chemical carcinogen DMBA (weighing 120 g) | - 6 months- 10mg TRF/kg/day<br>- gastric intubation | - Decreased LDL-C and TC levels induced by DMBA compared to normal control levels<br>- Decreased ALP activities and maintained low GST activities in liver | [71] |
| Tocotrienol | Bran | In vivo<br>- 4-week-old, HF diet treated mice | - 8 weeks<br>- HF diet supplemented with or without 10 mg of tocotrienols and 5% bran | - Co-treatment with bran and tocotrienols significantly supressed bodyweight gain in HFD-fed mice | [72] |

Table 3. *Cont.*

| Bioactive Compounds | Food Source | Study Subject or Model | Treatment or Intervention | Results | References |
|---|---|---|---|---|---|
| TRF | Rice bran | Human study<br>- RCT, parallel<br>- Hypercholesterolemic adults<br>- males <50 years and females <40 years) | - 35 days<br>- AHA Step-1 diet + TRF of rice bran 25, 50, 100, 200 mg/day | - Dose of 100 mg/day of TRF decreased serum TC, LDL, TG, and apolipoprotein B | [14] |
| Tocotrienol | RBO | Human study<br>- RCT, crossover<br>- T2D with hyperlipidemia<br>- mean age 48.5–52.6 years | - 60 days<br>- TRF capsules (2 capsules, 3 mg TRF/kg bw) vs. placebo (100 mg of TRF-free RBO/kg bw)<br>- Group A: TRF in the first phase and color matched placebo capsules in the second phase<br>- Group B: TRF in both first and second phases | - TRF supplement showed reduction of 23 to 42% in serum lipid | [73] |
| α-tocopherol and torotrienol | | | | | |
| α-tocopherol and torotrienol | Rice bran | In vivo<br>- 6-week-old, F344/slc male rats | - 21 days<br>- Group 1, normal chow diet; Group 2, WD; Group 3, WD with α-Toc 50 mg/day; Group 4, WD with rice bran tocotrienol 11.1 mg/day; Group 5, WD with α-tocopherol 50 mg/day and rice bran tocotrienol 11.1 mg/day<br>- oral administration | - α-tocopherol attenuated lowering effects of TC and TG rice bran tocotrienol<br>- α-tocopherol alone did not attenuate hyperlipidemic effects<br>- Rice bran tocotrienol-induced gene expression of CPT-1a and Cyp7a1 reduced by α-tocopherol | [78] |
| α-tocopherol and torotrienol | Rice bran | In vivo<br>- C57BL/6 ApoE-deficient mice | - 24 weeks<br>- α-tocopherol, TRF, didesmethyl tocotrienol (d-P25-tocotrienol) with low or high fat diets | - Reduced TC and LDL-C in TRF25 and d-P25-T3 group<br>- Reduced atherosclerotic lesion size | [79] |
| α-tocopherol and torotrienol | Rice bran | In vivo<br>- 6-week-old, KKAy diabetic mice | - 6 weeks<br>- normal diet (DM group), a diet including 0.1% Ricetrienol (RT group), non-diabetic C57BL mice (C group)<br>- 0.1% Ricetrienol (crude lipophilic rice bran extract) contains α-tocopherol, tocotrienol, phytosterol | - Rice bran treatment did not alter bodyweight, lipid levels, or glucose metabolism<br>- Elevation of the level of MDA in plasma significantly attenuated by rice bran<br>- The level of α-tocopherol in plasma of RT group was significantly higher compared to that in DM group | [80] |

AHA, American Heart Association; ALP, alkaline phosphatase; DM, diabetes mellitus; DMBA, 7,12-dimethylbenz [alpha]anthracene; GST, glutathione-S-transferase; HBA1c, percentage of glycosylated hemoglobin; HF, high fat; MDA, malondialdehyde; HMG-CoA, β-hydroxy β-methylglutaryl-CoA; NO, nitric oxide; PO, palm oil; RBO, rice bran oil; RCT, randomized, controlled clinical trial; T2D, Type 2 diabetes; TBARS, thiobarbituric acid reactive substance; TGF-β, transforming growth factor-beta; TRF, tocotrienol rich fraction; SOD, superoxide dismutase; STZ, streptozotocin; WD, Western diet.

### 5. Associations of Fiber with Cardiovascular and Metabolic Diseases

Dietary fiber is a carbohydrate polymer in plants that is not decomposed by the human digestive enzymes [81,82]. It helps to improve gut health by facilitating the excretion of feces and producing metabolites that are fermented by the gut microbiota and are beneficial to the gut environment [81,83]. In addition, it is known to improve various metabolic and cardiovascular diseases by suppressing the absorption of glucose and cholesterol, thereby reducing blood glucose, blood lipids, and body weight [84–88]. Whole grains also contain high levels of dietary fiber. A meta-analysis of various cohort studies has shown that the risk ratio for T2D is lower for cereal fiber than for fruit fiber, proving the superiority of whole grain dietary fiber [89]. In addition, the FDA has approved the claim that the consumption of soluble fiber, derived from whole grain oats, may decrease the risk of heart diseases when combined with a low saturated fat and cholesterol diet [90].

*β-Glucan*

The representative dietary fiber present in the whole grain is β-glucan [(1→3)(1→4)-β-D-glucan] and β-Glucan is a water-soluble fiber. As a specific indicator compound with various activities in oats and barley, its effects and mechanisms on metabolic and cardiovascular diseases have been studied [91].

First, oat and barley β-glucan effectively reduce blood lipid levels. Several human studies have reported that dietary fiber intake reduces blood TC and LDL-C levels [92–94]. As the cholesterol precursor lathosterol and the phytosterol campesterol decrease, it could be hypothesized that dietary fiber inhibits cholesterol synthesis and absorption [94]. In addition, β-glucan effectively inhibits lipid accumulation and absorption by reducing the intestinal uptake of fatty acids and inhibiting fatty acid synthesis pathways, such as ACC and FAS, as well as the expression of transport proteins, such as intestinal fatty acid binding protein (i-FABP) and fatty acid transport protein 4 (FATP4), as determined by preclinical studies [13].

Studies have also shown that β-glucan reduces blood glucose levels and improves insulin resistance. In animal studies, the administration of oat β-glucan reduced blood glucose, insulin, and fructosamine and improved glucose intolerance and insulin resistance [95,96]. Similarly, in clinical trials, in subjects with metabolic risk factors, oat β-glucan intake improved glucose intolerance and insulin sensitivity [92,97]. This is expected to be mainly due to the inhibition of glucose absorption by β-glucan. Specifically, in a diabetic animal model, the disaccharidase activity of the small intestinal mucosa is reduced, thereby delaying glucose digestion and, consequently, suppressing the rapid increase in blood glucose [95]. In addition, β-glucan, a representative soluble fiber, increases the viscosity of gastric contents, delaying the overall digestive process, including gastric emptying, thereby suppressing insulin secretion and increasing postprandial blood glucose [85,98] (Table 4).

**Table 4.** Effects of fiber on cardiovascular and metabolic diseases.

| Bioactive Compounds | Food Source | Study Subject or Model | Treatment or Intervention | Results | References |
|---|---|---|---|---|---|
| | | β-glucan | | | |
| β-glucan | Whole grain oats | Human study<br>- RCT, crossover<br>- Mildly hypercholesterolemic subjects<br>- 18–65 years | - 4 weeks<br>- β-glucan 5 g | - Decreased TC, LDL-C, TC/HDL-C ratio<br>- Decreased lathosterol and campesterol | [94] |
| β-glucan | Oat gum (80% β-glucan) | Human study<br>- RCT, crossover<br>- Hypercholesterolemic subjects | - 4 weeks<br>- oat gum (2.9 g β-glucan) | - Decreased TC and LDL-C | [93] |
| β-glucan | Whole grain oats and barley | In vitro<br>- Rat tissue (small intestine) | - 0.5% high-purity (HP)/high-viscosity (HV) barley, HP/low-viscosity (LV) barley, HP/HV oat, HP/LV oat, medium-purity (MP)/HV barley | - Decreased intestinal uptake of long chain fatty acid<br>- Decreased expression of ACC, FAS, i-FABP, and FATP4 | [13] |
| β-glucan | Whole grain oats | Human study<br>- RCT, parallel<br>- Hypercholesterolemic subjects<br>- 18–70 years | - 3 weeks<br>- β-glucan 5 mg | - Decreased TC<br>- Decreased postprandial glucose and insulin | [92] |
| β-glucan | Whole grain oats | In vivo<br>- SD rats (weighing 150–170 g) | - 4 weeks<br>- β-glucan 312.5 mg/kg<br>- oral administration | - Decreased bodyweight gain<br>- Decreased fasting blood glucose<br>- Increased ISI and NEFA<br>- Increased portion of *Bifidobacterium* and *Lactobacillus* in colon contents | [96] |
| β-glucan | Whole grain oats | In vivo<br>- 4-week-old, HSHF diet and STZ-induced diabetic mice | - 6 weeks<br>- β-glucan 2000, 1200, or 800 mg/kg<br>- oral administration | - Decreased fasting glucose and fructosamine<br>- Increased fasting insulin and decreased IAI<br>- Decreased activity of disaccharidase (sucrose, lactase, maltase) in small intestine mucosa | [95] |
| β-glucan | Whole grain oat | Human study<br>- RCT, parallel<br>- Subjects with elevated blood pressure<br>- ≥40 years | - 12 weeks<br>- Foods containing oat β-glucan (7.7 g/serving) | - Decreased SBP in subjects with BMI above the median (31.5 kg/m$^2$)<br>- Decreased glucose AUC, fasting insulin, peak insulin, and insulin AUC | [97] |

ACC, acetyl-CoA carboxylase; AUC, area under the curve; BMI, body mass index; FAS, fatty acid synthase; HDL-C, high-density lipoprotein cholesterol; HSHF, high-sucrose and high-fat diet; IAI, insulin activity index; i-FABP, intestinal fatty acid binding protein; FATP, fatty acid transport protein 4; ISI, insulin sensitivity index; LDL-C, low-density lipoprotein cholesterol; NEFA, non-esterified fatty acids; RCT, randomized, controlled clinical trial; SBP, systolic blood pressure; SCFA, short-chain fatty acid; SD, Sprague-Dawley; STZ, streptozotocin; TC, total cholesterol; TG, triglyceride.

## 6. Conclusions

Major bioactive compounds abundant in whole grains, such as polyphenols, flavonoids, antioxidants, and fiber, have been demonstrated to have preventive effects on cardiovascular and metabolic diseases. Alkylresorcinols, aventhramides, ferulic acid, and γ-oryzanol are representative polyphenols in whole grains, especially in rice bran or oat, and show preventive effects against impaired glucose intolerance, diabetes, vascular diseases, hypertension, and dyslipidemia. Rutin, one of the types a flavonoid rich in buckwheat, improved insulin sensitivity, glucose tolerance, lipid metabolism, and hypertension-related parameters. Tocotrienol, which are antioxidants that are generally derived from RBO and α-tocopherol, found in a variety of foods, also show hypolipidemic and hypoglycemic effects in vivo and in humans. In particular, the anti-hyperlipidemic effects of tocotrienol are attenuated by interactions with α-tocopherol. Soluble fiber, including β-glucan from oat and barley, decreased lipid profiles by inhibition of cholesterol or fatty acid synthesis and absorption, as well as blood glucose levels, with improved glucose intolerance and insulin resistance.

As described above, the preventive effects of bioactive compounds derived from whole grains on cardiovascular and metabolic diseases are well-established; therefore, whole grains are often consumed as a healthy food. Indeed, various studies have revealed that whole grain food consumption improves metabolic parameters. In a meta-analysis, intake of 3 servings, or 90 g, of whole grains per day was related to 19% and 22% reduced risks of coronary artery disease and cardiovascular diseases, respectively [25]. In addition, an overall inverse association between whole grain and metabolic syndrome has been identified, while a positive association has been detected for refined grain consumption in a meta-analysis of several observational studies [99].

The favorable effects of whole grain foods on cardiovascular and metabolic diseases have also been observed in clinical intervention studies. A higher consumption of a whole grain-enriched diet was associated with higher concentrations of lipid metabolites and decreased postprandial insulin levels compared to those for the intake of a refined wheat cereal diet in subjects who have metabolic syndrome [100]. Supplementation of a low-fiber diet with oat bran also improved non-HDL-C and TC compared to levels for the control diet [101]. The consumption of tartary buckwheat, a whole grain food, decreased the levels of insulin, LDL-C, and TC in patients who have T2D [102]. However, a non-significant association was observed in another study. Whole grain consumption did not alter metabolic disease parameters, anthropometric measurements, or body composition in those who were healthy, overweight, or obese and consumed whole grain wheat for 8 weeks compared to parameter values for the refined wheat product consumption group [103]. However, the amount of intake and study duration varies by studies. In addition, specific doses of bioactive compounds derived from whole grain foods and appropriate treatment duration of the studies described in this review are also still inconclusive. Thus, more studies of the relationship between the intake of bioactive compounds or whole grains and metabolic and cardiovascular diseases for specific guidelines should be conducted.

Taken together, these results indicate that bioactive compounds rich in whole grains, including alkylresorcinols, avenanthramides, ferulic acid, γ-oryzanol, rutin, tocotrienol, α-tocopherol, and β-glucan, as well as whole grains as a whole food, play important roles in the prevention of cardiovascular and metabolic diseases. However, data for the effects of individual bioactive compounds from well-designed human studies are still lacking, and the interactive effects of various bioactive compounds on the regulation of cardiovascular and metabolic diseases are still unclear. Since plant-derived components are widely used as new nutritional and pharmaceutical supplements, owing to their minimal side effects, further studies are required. In conclusion, the intake of whole grain-derived bioactive compounds, for cardiovascular and metabolic disease prevention, may be a useful dietary strategy.

**Author Contributions:** Conceptualization, S.C. and S.-H.P.; Methodology, S.C. and S.-H.P.; Investigation, S.C., J.-T.H. and S.-H.P.; Data Curation, S.C., J.-T.H. and S.-H.P.; Writing—Original Draft Preparation, S.C.; Writing—Review & Editing, S.-H.P.; Supervision, S.-H.P. All authors have read and agreed to the published version of the manuscript.

**Funding:** This study was funded by a research grant from the Korea Food Research Institute (Project Number: E0210601), Republic of Korea.

**Institutional Review Board Statement:** Not applicable.

**Informed Consent Statement:** Not applicable.

**Data Availability Statement:** Not applicable.

**Acknowledgments:** This study was supported by a Korea Food Research Institute grant (Project Number: E0210601).

**Conflicts of Interest:** The authors declare no conflict of interest.

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
