# Peer review of "Physiological Effects of Bioactive Compounds Derived from Whole Grains on Cardiovascular and Metabolic Diseases"

_applsci, doi:10.3390/app12020658_

Round 1
Reviewer 1 Report
This review paper aimed to evaluate the characteristics and functions of active ingredients in whole grains and the effects of whole-grain intake on metabolic and cardiovascular diseases. The manuscript was well written by authors who conducted plenty of reviewing work related to the topic. The topic coincides with the scope of the journal. However, slight improvement is also needed.
The logic is very clear that the manuscript has summarized the association of different bioactive ingredients of whole grains with cardiovascular and metabolic diseases from four specified aspects, i.e. polyphenols, flavonoids, antioxidants, and dietary fiber, while it would be better if there were more information in the ‘1. Introduction (Background)’ to briefly introduce the inner connection of whole-grain and aforementioned each aspect of bioactive ingredients, and explain why summarizing the effects of whole-grain from those four aspects, but the other aspects.
It was obvious the whole grain consumption might alter glucose intolerance, insulin resistance, cholesterol levels, etc., and consequently affected cardiovascular and metabolic diseases, but the effects depend on the volume per day and duration of consumption (from weeks to years). Thus, the review paper would be more meaningful if a summarized recommendation including suggested treatment volume and duration had been provided in the ‘6. Conclusion’, instead of concluding: ‘more studies of the association between the intake of bioactive compounds or whole grains and metabolic and cardiovascular diseases should be conducted.’ (Page 25, line 164-165)
Please provide references to some concepts:
Page 1, line 34-36, line 38-39, line 42-44; Page 2, line 68-70
Author Response
This review paper aimed to evaluate the characteristics and functions of active ingredients in whole grains and the effects of whole-grain intake on metabolic and cardiovascular diseases. The manuscript was well written by authors who conducted plenty of reviewing work related to the topic. The topic coincides with the scope of the journal. However, slight improvement is also needed.
The logic is very clear that the manuscript has summarized the association of different bioactive ingredients of whole grains with cardiovascular and metabolic diseases from four specified aspects, i.e. polyphenols, flavonoids, antioxidants, and dietary fiber, while it would be better if there were more information in the ‘1. Introduction (Background)’ to briefly introduce the inner connection of whole-grain and aforementioned each aspect of bioactive ingredients, and explain why summarizing the effects of whole-grain from those four aspects, but the other aspects.
--> As you commented, we described inner connection of whole grain and bioactive ingredients as follows:
p. 2, line 57-70: Representatively, there are phenol- and dietary fiber- derived compounds. In addition, it contains various tocols, which are vitamin E with strong antioxidant activity. They are known to be involved in blood glucose and lipid metabolism as well as antioxidant and anti-inflammatory effects [29-21]. Due to the influence of these various bioactive compounds, it has been reported that whole grain consumption is the main factor affecting mortality due to cardiovascular disease [23]. … However, the associations between various bioactive compounds in whole grains and risk factors for metabolic and cardiovascular diseases have rarely been reviewed.
p. 2, line 78-84: In particular, we describe recent findings on phenol-, vitamin E-, and dietary fiber-derived bioactive compounds, which are representative bioactive compounds in the following four major groups (including eight compounds): whole grain-specific polyphenols, including alkylresorcinols (ARs), avenanthramides (Avns), ferulic acids (FA), and γ-oryzanol (OZ); flavonoids, including rutin; vitamin E, including tocotrienol and α-tocopherol; and dietary fiber, including β-glucan.
It was obvious the whole grain consumption might alter glucose intolerance, insulin resistance, cholesterol levels, etc., and consequently affected cardiovascular and metabolic diseases, but the effects depend on the volume per day and duration of consumption (from weeks to years). Thus, the review paper would be more meaningful if a summarized recommendation including suggested treatment volume and duration had been provided in the ‘6. Conclusion’, instead of concluding: ‘more studies of the association between the intake of bioactive compounds or whole grains and metabolic and cardiovascular diseases should be conducted.’ (Page 25, line 164-165)
--> As you commented, we revised conclusion as follows:
p. 26, line 166-167: However, the amount of intake and study duration varies by studies. In addition, specific doses of bioactive compounds derived from whole grain foods and appropriate treatment duration of the studies described in this review are also still inconclusive. Thus, more studies of the association between the intake of bioactive compounds or whole grains and metabolic and cardiovascular diseases for specific guidelines should be conducted.
Please provide references to some concepts:
Page 1, line 34-36, line 38-39, line 42-44; Page 2, line 68-70
--> As commented, we added references as follows:
p. 1, line 35-37
3. Olatona, F.A.; Onabanjo, O.O.; Ugbaja, R.N.; Nnoaham, K.E.; Adelekan, D.A. Dietary Habits and Metabolic Risk Factors for Non-Communicable Diseases in a University Undergraduate Population. J. Health Popul. Nutr. 2018, 37 (1), 21.
4. Grundy, S.M. Metabolic Syndrome Pandemic. Arterioscler. Thromb. Vasc. Biol. 2008, 28 (4), 629-636.
p. 1, line 42-45
7. Trumbo, P.; Schlicker, S.; Yates, A.A.; Poos, M.; Food, Nutrition Board of the Institute of Medicine TNA: Dietary Reference Intakes for Energy, Carbohydrate, Fiber, Fat, Fatty Acids, Cholesterol, Protein and Amino Acids. J. Am. Diet Assoc. 2002, 102 (11), 1621-1630.
8. Jiang, H.; Zhang, J.; Du, W.; Su, C.; Zhang, B.; Wang, H.; Energy Intake and Energy Contributions of Macronutrients and Major Food Sources Among Chinese Adults: CHNS 2015 and CNTCS 2015. Eur. J. Clin. Nutr. 2021, 75 (2), 314-324.
9. Gose, M.; Krems, C.; Heuer, T.; Hoffmann, I. Trends in Food Consumption and Nutrient Intake in Germany between 2006 and 2012: Results of the German National Nutrition Monitoring (NEMONIT). Br. J. Nutr. 2016, 115 (8),1498-1507.
10. Wright, J. D., Wang, C.Y.: Trends in Intake of Energy and Macronutrients in Adults from 1999-2000 through 2007-2008. NCHS Data Brief. 2010 (49), 1-8.
p. 2, line 94-96
29. Baerson, S.R.; Schröder, J.; Cook, D.; Rimando, A.M.; Pan, Z.; Dayan, F.E.; Noonan, B.P.; Duke, S.O. Alkylresorcinol Biosynthesis in Plants: New Insights from an Ancient Enzyme Family? Plant Signal. Behav. 2010, 5 (10), 1286–1289.
30. Quistad, G.B.; Staiger, L.E.; Schooley, D.A. Environmental Degradation of the Miticide Cycloprate (Hexadecyl Cyclopro-panecarboxylate). 1. Rat Metabolism. J. Agric. Food Chem. 1978, 26 (1), 60–66.
31. Zhao, Z.; Moghadasian, M.H. Chemistry, Natural Sources, Dietary Intake and Pharmacokinetic Properties of Ferulic Acid: A Review. Food Chem. 2008, 109 (4), 691–702.
32. Scavariello, E.M.; Arellano, D.B. [Gamma-Oryzanol: an Important Component in Rice Brain Oil]. Arch. Latinoam. Nutr. 1998, 48 (1), 7–12.

Reviewer 2 Report
Interesting review to raise up on what is an important grain on both metabolic disease and CVD's.
However, some parts need to be improved and clarified. Attached of my comments in the pdf file.

Author Response
Interesting review to raise up on what is an important grain on both metabolic disease and CVD's.
However, some parts need to be improved and clarified. Attached of my comments in the pdf file.
--> Thank you for your comments. We revised several sentences and expressions in Tables which you marked in yellow because we couldn't find your specific comments. If the correction is not appropriate, please give us more comments.
p. 1, Abstract (line 19, 24): The germ and bran of grains are rich in compounds, including phytochemicals, vitamins, minerals, and dietary fiber, and these compounds are effective in preventing and improving metabolic and cardiovascular diseases. Thus, this review describes the characteristics and functions of bioactive ingredients in whole grains, focusing on mechanisms by which polyphenols, antioxidants, and dietary fiber contribute to cardiovascular and metabolic diseases based on preclinical and clinical studies. There is clear evidence for the broad preventive and therapeutic effects of whole grains, supporting the value of early dietary intervention.
p. 6, Table 1; p. 15, Table 2; p. 19, Table 3; p. 24, Table 4: Bioactive compounds, Food source, Study subject or model, Treatment or intervention, Results, References

Reviewer 3 Report
The manuscript by Chung and colleagues provides a comprehensive review of the beneficial effects of the major bioactive compounds present in whole grains in cardiovascular and metabolic diseases. Overall the review provides thorough information from in vitro and in vivo studies, the latter coming from both animal and clinical studies.
In terms of structure the manuscript is well-written and easy to read, although the syntax could be improved in some paragraphs; the tables provide a useful complement to the information of the text. However, the manuscript would benefit from the introduction of some figures, namely (1) of the constitution of whole grains (please check: https://www.mdpi.com/2304-8158/10/8/1765/htm), and (2) of the main structure (or examples) of the major bioactive compounds discussed in the manuscript.
In terms of content, the manuscript would benefit from the discussion of the potentially negative effects of whole grains consumption, so as to prevent presenting a skewed revision. I fail to understand the logic behind the authors’ decision to group the discussed compounds as “polyphenols”, “flavonoids” and “antioxidants”. One the one hand flavonoids are a major group of polyphenols; on the other hand several polyphenols and other compounds show antioxidant activity. As such, the authors should better justify their choice in making such a division which clearly does not distinguish between structure and function of the discussed compounds. Besides this, I identify several other issues that need revision/correction:
Lines (L) 30-31: The sentence needs to be improved, as in its present state it implies that hypertension is either a risk factor for metabolic disease or a metabolic disease itself, when that is not the case;
L33: maybe it would make more sense to add that lowering the prevalence of such diseases is a major challenge in present day;
L41: Please provide examples of whole grains, similarly to the abstract;
L42: By “people” do the authors mean in general, irrespective of culture and/or geography? Please clarify. Also, please provide a reference to support the information provided in this sentence;
L42: By “energy” do the authors mean “daily energy intake”? Please clarify;
L54: Are there more meta-analysis providing similar or complementing information? If so, it would be useful to mention more studies in order to provide a broader view of the link between whole grain consumption and cardiovascular risk;
L75 and similar cases: in formal English sentences do not begin with accronyms;
L82: it should be “ARs”;
L89: the result of the study pertaining to ischemic stroke was introduced right after a lengthy paragraph focused solely on metabolic disease. As such, I would suggest adding a sentence before this one to make a smoother change from one type of disease to the next;
L97: keeping with the authors’ preference throughout the text, it is better to write “LDLr-/-“ (please check also Table 1);
L102: it should be “cyclic guanosine”;
L104: “human aortic endothelial cells” should appear before “SMCs”;
L119: I suggest writing “phenolic acid” instead of “phenolic compound”;
L119: It should be clearer in the text that that “gamma-oryzanol” is a group of compounds;
L126: did FA and OZ decrease HDL-C levels? Please double-check!;
L130: please erase “)” after “vitamin E”;
L137: It should be “Furthermore”;
L149: instead of “involved in” I suggest writing “responsible for”;
L155: which types of vessels? Arteries, veins, both?;
L157: perhaps the word “effects” is missing after “pressure-lowering”;
L158: In the “Food” column, the specific whole grain must be mentioned whenever possible, as well as the mean age of the subjects (animals, humans) in the “Model” column (this also applies to Tables 2-4); the “Exposure” cells lack information – the posology of a given compound should be clearly stated (this also applies to Tables 2-4); also, I suggest adding “homologs” after “AR C17 and C19”; the cell in the “Exposure” column pertaining to “DHPPA” is empty, please double-check if this is intended; in the second study pertaining to “FA and OZ”, the term “hydroperoxide” should be referred to in plural “hydroperoxides”; in the third “FA” study, the acronym “HUVEC” should appear in plural “HUVECs”; a definition of “Plant materials” should be given in the text; the “2” in “thromboxane B2” should appear in subscript;
L184-186: the sentence lacks a reference. Also, this sentence should appear in the next paragraph, right next to the other sentences discussing similar results;
L204: instead of “a well-known” I suggest “the well-known”;
L41-42 of section 4: please define “ALP” and “GST”;
L59 of section 4: what do the authors mean by “lipophilicity” in this context? Did they mean “fluidity”?;
L65 of section 4: the names of the genes are not capitalized;
L75 of section 4: the “Food” cell of the first “TRF” study is empty; please double-check if this is intended;
L82 of section 5: “in plants” should appear right after “polymer”;
L94-95 of section 5: please consider merging the first two sentences;
L133 of section 5: please revise the sentence beginning with “Flavonoids and rutin”, it is confusing;
L136-137 of section 5: these terms are inconsistent with the ones present throughout the text (i.e., hypoglycemic, hypolipidemic), please uniformize;
Author Response
The manuscript by Chung and colleagues provides a comprehensive review of the beneficial effects of the major bioactive compounds present in whole grains in cardiovascular and metabolic diseases. Overall the review provides thorough information from in vitro and in vivo studies, the latter coming from both animal and clinical studies.
In terms of structure the manuscript is well-written and easy to read, although the syntax could be improved in some paragraphs; the tables provide a useful complement to the information of the text. However, the manuscript would benefit from the introduction of some figures, namely (1) of the constitution of whole grains (please check: https://www.mdpi.com/2304-8158/10/8/1765/htm), and (2) of the main structure (or examples) of the major bioactive compounds discussed in the manuscript.
--> As you commented, we added figures as follows:
p. 1: Figure 1. Whole grain wheat composition.
p. 2: Figure 2. Main structure of whole grain-derived bioactive compounds (a) Alkylresorcinol, (b) Avenanthramide, (c) ferulic acid, (d) γ-oryzanol, (e) rutin, (f) tocotrienol, (g) α-tocopherol, (h) β-glucan
In terms of content, the manuscript would benefit from the discussion of the potentially negative effects of whole grains consumption, so as to prevent presenting a skewed revision.
--> Numerous studies have been conducted to discuss the health benefits of whole grains. However, since a careful approach is required for the intake of single bioactive compounds contained in whole grains, comments on the intake of bioactive compounds are explained as follows:
p. 26, line 166-167: However, the amount of intake and study duration varies by studies. In addition, specific doses of bioactive compounds derived from whole grain foods and appropriate treatment duration of the studies described in this review are also still inconclusive. Thus, more studies of the association between the intake of bioactive compounds or whole grains and metabolic and cardiovascular diseases for specific guidelines should be conducted.
I fail to understand the logic behind the authors’ decision to group the discussed compounds as “polyphenols”, “flavonoids” and “antioxidants”. One the one hand flavonoids are a major group of polyphenols; on the other hand several polyphenols and other compounds show antioxidant activity. As such, the authors should better justify their choice in making such a division which clearly does not distinguish between structure and function of the discussed compounds.
--> As you pointed out, since the classification of the major groups was unclear, we added flavonoids to the polyphenol group as follows:
p. 2, line 57-60: Representatively, there are phenol- and dietary fiber-derived compounds. In addition, it contains various tocols, which are vitamin E with strong antioxidant activity. They are known to be involved in blood glucose and lipid metabolism as well as antioxidant and anti-inflammatory effects [19-22]. Due to the influence of these various bioactive compounds, it has been reported that whole grain consumption is the main factor affecting mortality due to cardiovascular disease [23].
p. 2, line 76-84: Therefore, a literature review was performed to evaluate the characteristics and functions of active ingredients in whole grains and the effects of whole grain intake on metabolic and cardiovascular diseases. In particular, we describe recent findings on phenol-, vitamin E-, and dietary fiber-derived bioactive compounds, which are representative bioactive compounds in the following four major groups (including eight compounds): whole grain-specific polyphenols, including alkylresorcinols (ARs), avenanthramides (Avns), ferulic acids (FA), and γ-oryzanol (OZ); flavonoids, including rutin; vitamin E, including tocotrienol and α-tocopherol; and dietary fiber, including β-glucan (Figure 2).
p. 3, line 89: 2. Association of whole grain-specific polyphenols with cardiovascular and metabolic diseases
--> In addition, the compounds of the “antioxidants” part belongs to the vitamin E, so the subtitle modified to vitamin E as follows:
p. 17, line 1: Associations of vitamin E with cardiovascular and metabolic diseases
Besides this, I identify several other issues that need revision/correction:
Lines (L) 30-31: The sentence needs to be improved, as in its present state it implies that hypertension is either a risk factor for metabolic disease or a metabolic disease itself, when that is not the case;
--> As you commented, we deleted “hypertension” in the sentence to avoid confusion (p. 1, line 32).
L33: maybe it would make more sense to add that lowering the prevalence of such diseases is a major challenge in present day;
--> As you commented, we revised “medical science” to “today” (p. 1, line 34).
L41: Please provide examples of whole grains, similarly to the abstract;
--> As you commented, we added examples of whole grains (e.g., oats, barley, and buckwheat) (p.1, line 42-43).
L42: By “people” do the authors mean in general, irrespective of culture and/or geography? Please clarify. Also, please provide a reference to support the information provided in this sentence;
-->We considered “people (general)” were people with average eating habits around the world, and as you pointed out, we revised it to be more specific as follows:
p. 1, line 43-44: In general, people around the world, on average, get almost 50% of their daily energy intake from carbohydrates [7-10].”
--> We also added references as follows:
7. Trumbo, P.; Schlicker, S.; Yates, A.A.; Poos, M.; Food, Nutrition Board of the Institute of Medicine TNA: Dietary Reference Intakes for Energy, Carbohydrate, Fiber, Fat, Fatty Acids, Cholesterol, Protein and Amino Acids. J. Am. Diet Assoc. 2002, 102 (11), 1621-1630.
8. Jiang, H.; Zhang, J.; Du, W.; Su, C.; Zhang, B.; Wang, H.; Energy Intake and Energy Contributions of Macronutrients and Major Food Sources Among Chinese Adults: CHNS 2015 and CNTCS 2015. Eur. J. Clin. Nutr. 2021, 75 (2), 314-324.
9. Gose, M.; Krems, C.; Heuer, T.; Hoffmann, I. Trends in Food Consumption and Nutrient Intake in Germany between 2006 and 2012: Results of the German National Nutrition Monitoring (NEMONIT). Br. J. Nutr. 2016, 115 (8),1498-1507.
10. Wright, J. D., Wang, C.Y.: Trends in Intake of Energy and Macronutrients in Adults from 1999-2000 through 2007-2008. NCHS Data Brief. 2010 (49), 1-8.
L42: By “energy” do the authors mean “daily energy intake”? Please clarify;
--> As you commented, we revised it (p. 1, line 44).
L54: Are there more meta-analysis providing similar or complementing information? If so, it would be useful to mention more studies in order to provide a broader view of the link between whole grain consumption and cardiovascular risk;
--> As you commented, we further described another meta-analysis of the effect of whole grain as follows:
p. 2, line 64-66: In another meta-analysis study with various cohorts, it was also reported that consuming 3 servings of whole grains reduced the relative risk of mortality from cancers and metabolic diseases, as well as cardiovascular disease prevalence [25].
L75 and similar cases: in formal English sentences do not begin with acronyms;
--> As you commented, we revised it (p.3, line 98, 123, 149).
L82: it should be “ARs”;
-->As you commented, we revised it (p.4, line 108).
L89: the result of the study pertaining to ischemic stroke was introduced right after a lengthy paragraph focused solely on metabolic disease. As such, I would suggest adding a sentence before this one to make a smoother change from one type of disease to the next;
--> As you commented, we added a phrase linking the aforementioned studies with the later mentioned study as follows:
p. 4, line 117-121: As such, ARs showed an excellent effect in controlling blood glucose and lipids that directly or indirectly affect cardiovascular diseases. Therefore, these study results provide scientific evidences to support a human study that elevated blood DHPPA levels induced by whole grain intake lowered the prevalence of ischemic stroke [16] (Table 1).
L97: keeping with the authors’ preference throughout the text, it is better to write “LDLr-/-“ (please check also Table 1);
--> As you commented, we revised it (p.4, line 127).
L102: it should be “cyclic guanosine”;
--> As you commented, we revised it (p. 4, line 132).
L104: “human aortic endothelial cells” should appear before “SMCs”;
--> As you commented, we revised it (p. 4, line 134-135).
L119: I suggest writing “phenolic acid” instead of “phenolic compound”;
--> As you commented, we revised it (p. 4, line 150).
L119: It should be clearer in the text that that “gamma-oryzanol” is a group of compounds;
--> As you commented, we revised the phrase to clarify as follows:
p. 4, line 150-151: γ-oryzanol is a ferulic acid mixture in which triterpenoids are bound to esterified ferulic acid.
L126: did FA and OZ decrease HDL-C levels? Please double-check!;
--> FA and OZ increase HDL-C levels. Therefore, the wrong part has been corrected as follows:
p. 5, line 157-159: FA and OZ decreased body weight, body fat, and blood lipid (TC, triglyceride [TG], low-density lipoprotein cholesterol [LDL-C], free fatty acid [FFA]) levels and increased high-density lipoprotein cholesterol (HDL-C).
L130: please erase “)” after “vitamin E”;
--> As you commented, we deleted it (p. 5, line 163).
L137: It should be “Furthermore”;
--> As you commented, we revised it (p. 5, line 170).
L149: instead of “involved in” I suggest writing “responsible for”;
--> As you commented, we revised it (p. 5, line 182).
L155: which types of vessels? Arteries, veins, both?;
--> It was arteries and we added a detailed description as follows:
p. 5, line 188: arterial blood vessel walls
L157: perhaps the word “effects” is missing after “pressure-lowering”;
--> As you commented, we added the word (p. 5, line 190).
L158: In the “Food” column, the specific whole grain must be mentioned whenever possible, as well as the mean age of the subjects (animals, humans) in the “Model” column (this also applies to Tables 2-4);
--> As you commented, mean age of the subjects and food sources of bioactive compounds are further described as far as possible (p. 6, Table 1; p. 15, Table 2; p. 19, Table 3; p. 24, Table 4).
the “Exposure” cells lack information – the posology of a given compound should be clearly stated (this also applies to Tables 2-4);
--> As you commented, we added method of administration/supplementation (p. 6, Table 1; p. 15, Table 2; p. 19, Table 3; p. 24, Table 4).
also, I suggest adding “homologs” after “AR C17 and C19”;
--> As you commented, we added the word [Table 1, first study (alkylresorcinols)].
the cell in the “Exposure” column pertaining to “DHPPA” is empty, please double-check if this is intended;
--> As you commented, we added it as follows:
Table 1, 6th study (alkylresorcinls): concentration of DHPPA
in the second study pertaining to “FA and OZ”, the term “hydroperoxide” should be referred to in plural “hydroperoxides”;
--> As you commented, we revised it [Table 1, second study (FA and OZ)].
in the third “FA” study, the acronym “HUVEC” should appear in plural “HUVECs”;
--> As you commented, we revised it[Table 1, 8th study (FA and OZ)].
a definition of “Plant materials” should be given in the text;
--> As you commented, we added a detailed description (line 71-72).
p. 3, line 92-93: Polyphenols are phenolic compounds with one or more phenol units per molecule and are present in most plant materials such as vegetables, fruits, and grains [26].
the “2” in “thromboxane B2” should appear in subscript;
--> As you commented, we revised it (p. 5, line 185).
L184-186: the sentence lacks a reference. Also, this sentence should appear in the next paragraph, right next to the other sentences discussing similar results;
--> As you commented, we revised sentences (p. 14, line 220-221).
L204: instead of “a well-known” I suggest “the well-known”;
--> As you commented, we revised the word (p.14, line 238).
L41-42 of section 4: please define “ALP” and “GST”;
--> As you commented, we defined “ALP and “GST” in the sentence as follows:
p. 17, line 41-42: In addition, TRF isolated from RBO at a dose of 10 mg/kg bw/day reduced TC and LDL-C by 30% and 67%, respectively, and decreased alkaline phosphatase (ALP) activity and maintained low glutathione-S-transferase (GST) activity…
L59 of section 4: what do the authors mean by “lipophilicity” in this context? Did they mean “fluidity”?;
--> We added detailed explanation of the “lipophilicity” in the sentence as follows:
p. 18, line 58-59: These structures confer lipophilicity, which means the property of a chemical components dissolve in lipids, to lipid bilayers of lipoproteins or membranes.
L65 of section 4: the names of the genes are not capitalized;
--> As you commented, we revised it (p. 18, line 66 and p. 21, Table 3).
L75 of section 4: the “Food” cell of the first “TRF” study is empty; please double-check if this is intended;
--> The food of “TRF” of the first study in Table 3 is not described in the article. We added “N/A” to avoid confusion [Table 3, first study (Tocotrienol)].
L82 of section 5: “in plants” should appear right after “polymer”;
--> As you commented, we revised it (p. 23, line 82).
L94-95 of section 5: please consider merging the first two sentences;
--> As you commented, we revised it (p. 23, line 96).
L133 of section 5: please revise the sentence beginning with “Flavonoids and rutin”, it is confusing;
--> We revised the word (section 6) according to your comment as follows:
p. 26, line 134: Rutin, one of types a flavonoid rich in buckwheat, improved insulin sensitivity, glucose tolerance, lipid metabolism, and hypertension-related parameters.
L136-137 of section 5: these terms are inconsistent with the ones present throughout the text (i.e., hypoglycemic, hypolipidemic), please uniformize;
--> We revised the word (section 6) according to your comment as follows:
p. 26, line 138: hypolipidemic and hypoglycemic effects

Round 2
Reviewer 3 Report
The authors have satisfactorily addressed all my concerns. I agree that this manuscript should be accepted for publication.
Author Response
Thank you for your review.
In addition, we have revised text to exclude the possibility of plagiarism.